# Biochemical reconstitution of branching microtubule nucleation

**Raymundo Alfaro-Aco[1†], Akanksha Thawani[2‡], Sabine Petry[1§*]**

[1]Department of Molecular Biology, Princeton University, Princeton, United States; [2]Department of Chemical and Biological Engineering, Princeton University, Princeton, United States

**Abstract** Microtubules are nucleated from specific locations at precise times in the cell cycle. However, the factors that constitute these microtubule nucleation pathways and their mode of action still need to be identified. Using purified *Xenopus laevis* proteins we biochemically reconstitute branching microtubule nucleation, which is critical for chromosome segregation. We found that besides the microtubule nucleator gamma-tubulin ring complex (γ-TuRC), the branching effectors augmin and TPX2 are required to efficiently nucleate microtubules from pre-existing microtubules. TPX2 has the unexpected capacity to directly recruit γ-TuRC as well as augmin, which in turn targets more γ-TuRC along the microtubule lattice. TPX2 and augmin enable γ-TuRC-dependent microtubule nucleation at preferred branching angles of less than 90 degrees from regularly-spaced patches along microtubules. This work provides a blueprint for other microtubule nucleation pathways and helps explain how microtubules are generated in the spindle.

**\*For correspondence:**
spetry@Princeton.EDU

**Present address:** [†]Department of Molecular Biology, Princeton University, Princeton, United States; [‡]Department of Chemical and Biological Engineering, Princeton University, Princeton, United States; [§]Department of Molecular Biology, Princeton University, Princeton, United States

**Competing interests:** The authors declare that no competing interests exist.

## Introduction

Microtubules are nucleated from specific locations in the cell, and several of these microtubule nucleation pathways converge to form a particular cytoskeletal architecture (*Kollman et al., 2011*; *Lin et al., 2015*; *Lüders and Stearns, 2007*). Importantly, microtubules in cells are nucleated by the microtubule nucleator γ-TuRC (*Kollman et al., 2011*; *Zheng et al., 1995*) and its co-nucleation factor XMAP215 (*Flor-Parra et al., 2018*; *Gunzelmann et al., 2018*; *Thawani et al., 2018*). At the same time, each microtubule nucleation pathway requires a unique set of nucleation effectors to recruit and regulate γ-TuRC at distinct cellular locations (*Lin et al., 2015*). The identity of most of these effectors remains elusive, along with a mechanistic understanding of how they constitute the different microtubule nucleation pathways that generate the cytoskeleton.

Microtubules can nucleate from pre-existing microtubules, termed branching microtubule nucleation (*Petry et al., 2013*), which amplifies microtubule number while preserving their polarity, as is needed in the mitotic spindle and in axons (*Cunha-Ferreira et al., 2018*; *David et al., 2019*; *Kamasaki et al., 2013*; *Petry et al., 2013*; *Sánchez-Huertas et al., 2016*). The eight-subunit protein complex augmin is required for branching microtubule nucleation in plant, human and *Drosophila* cells, and meiotic Xenopus egg extract, where its depletion leads to reduced spindle microtubule density, less kinetochore fiber tension, metaphase arrest, and cytokinesis failure (*David et al., 2019*; *Decker et al., 2018*; *Goshima et al., 2008*; *Hayward et al., 2014*; *Ho et al., 2011*; *Kamasaki et al., 2013*; *Lawo et al., 2009*; *Nakaoka et al., 2012*; *Petry et al., 2011*; *Uehara et al., 2009*). Augmin is necessary to recruit γ-TuRC to spindle microtubules (*Goshima et al., 2007*), and following the recombinant expression of augmin (*Hsia et al., 2014*), this activity was confirmed using purified proteins (*Song et al., 2018*).

In meiotic Xenopus egg extract, the Ran-regulated protein TPX2 is released near chromatin (*Gruss et al., 2001*), where it stimulates branching microtubule nucleation (*Petry et al., 2013*), potentially by activating γ-TuRC via nucleation activator motifs (*Alfaro-Aco et al., 2017*). Recently,

TPX2 was also observed to form a co-condensate with tubulin along the microtubule lattice, which enhances the kinetic efficiency of branching microtubule nucleation (*King and Petry, 2019*). In meiotic Xenopus egg extract, TPX2 needs to bind to microtubules before augmin/γ-TuRC to result in a successful nucleation event (*Thawani et al., 2019*). In contrast, in mitotic *Drosophila* cells TPX2 is not required, and augmin can bind to microtubules before γ-TuRC (*Verma and Maresca, 2019*). Despite these numerous studies to characterize each individual protein component, exactly how augmin, TPX2 and γ-TuRC together mediate branching microtubule nucleation, and whether they alone constitute a minimal system that nucleates branched microtubules, remains unclear. Here, we use biochemical reconstitution of its purified components to mechanistically dissect branching microtubule nucleation.

## Results and discussion

Branching microtubule nucleation has been studied in Xenopus egg extract, where it is elicited by the constitutively active version of Ran (RanQ69L) (*Petry et al., 2013*). In order to establish a controlled, minimal assay that furthers our mechanistic insight, we exposed a microtubule tethered to glass to sequential reaction mixtures of decreasing complexity and thereby regulated the availability of proteins necessary to stimulate branching microtubule nucleation. Using multicolor time-lapse total internal reflection (TIRF) microscopy, we first confirmed that an endogenous, pre-existing microtubule can serve as a template for branching microtubule nucleation when exposed to Ran-supplemented extract that releases branching factors (*Figure 1A* and *Figure 1—video 1*). This shows that a microtubule formed independent of Ran can serve as the site for binding of branching factors and subsequent nucleation events.

To gain mechanistic insight, we hypothesized that all necessary Ran-regulated branching factors bind to the pre-existing microtubule prior to microtubule nucleation. To test this, Ran-regulated branching factors were allowed to bind to taxol-stabilized pre-existing microtubules in the presence of nocodazole, which inhibits new microtubule formation (*Figure 1B* and *Figure 1—figure supplement 1*). When another extract reaction was subsequently added, new microtubules nucleated almost exclusively from pre-existing microtubules, indicating that Ran-regulated branching factors bind to microtubules independent of their successful nucleation reactions (*Figure 1B*). Importantly, when RanQ69L was omitted and no branching factors were released in the second extract reaction, pre-existing microtubules simply elongated and branching microtubule nucleation was absent in the third reaction (*Figure 1—figure supplement 2A*).

To further test whether these microtubule-bound branching factors are sufficient for generating branches, we reduced the complexity of our assay by introducing only purified tubulin and GTP in the third reaction step. Surprisingly, short branched microtubules nucleated from pre-existing microtubules (*Figure 1—figure supplement 2B*), showing that upon binding of branching factors and γ-TuRC, tubulin is the only protein required to form new branched microtubules from the localized factors. Further addition of XMAP215 to the final tubulin reaction made the short branches grow longer (*Figure 1C*). This revealed that branched microtubules retained the polarity of the pre-existing microtubule, and new microtubules do not appear to nucleate from other branched microtubules, suggesting that the branching factors do not relocate between microtubules (*Figure 1C* and *Figure 1—video 2*). Thus, solely the deposition of branching factors and γ-TuRC to the pre-existing microtubule determines branching architecture.

Because the key for branching microtubule nucleation is to target γ-TuRC along the length of a microtubule, we verified as a proof of concept that purified γ-TuRC, when tethered along the microtubule lattice via artificial linkers, can still nucleate microtubules, which indeed was the case (*Figure 2—figure supplement 1A–B*). Therefore, if all branching factors are known, it should be possible to reconstitute branching microtubule nucleation from a template microtubule using only purified components. To test this, we purified the essential proteins for branching microtubule nucleation in Xenopus egg extract (*Petry et al., 2013*). The GFP-labeled eight-subunit *X. laevis* augmin holocomplex was co-expressed in insect cells and co-purified (*Song et al., 2018*), the native 2.2 MDa γ-TuRC was purified from Xenopus egg extract (*Thawani et al., 2018*), and GFP-TPX2 was expressed from *E. coli* and purified (*King and Petry, 2019*) (*Figure 2—figure supplement 2A–C*).

First, we assessed how the nucleator γ-TuRC gets targeted along the microtubule lattice. Purified TPX2, augmin and γ-TuRC were added in various combinations to surface-bound, GMPCPP-

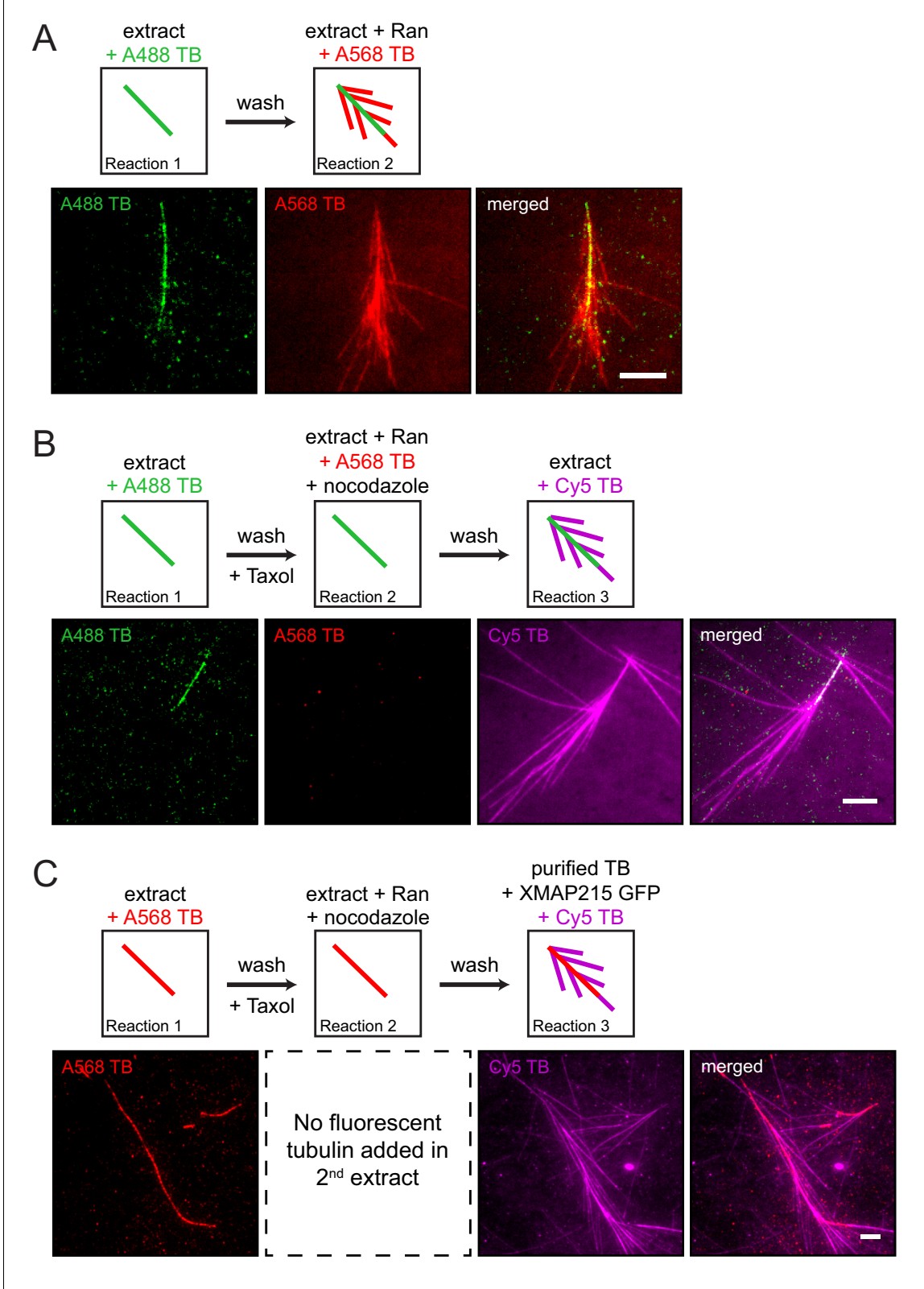

**Figure 1.** The proteins necessary for branching microtubule nucleation in Xenopus egg extract bind to a pre-existing microtubule independent of the nucleation event. (**A–C**) Sequential reactions with Xenopus egg extract. (**A**) Single microtubules formed on the glass surface in the first extract supplemented with Alexa488 tubulin (green). A second extract supplemented with Alexa568 tubulin (red) and RanQ69L was subsequently introduced. New microtubules (red) nucleated from pre-existing microtubules (green). See *Figure 1—video 1*. (**B**) Single microtubules formed on the glass surface

*Figure 1 continued on next page*

*Figure 1 continued*

in the first extract supplemented with Alexa488 tubulin (green). A second extract supplemented with Alexa568 tubulin (red), RanQ69L and nocodazole was subsequently introduced, followed by a third extract supplemented with Cy5 tubulin (magenta). Branched microtubules (magenta) nucleated from pre-existing microtubules (green) via the branching factors released in the second extract, while no microtubules formed in the presence of nocodazole (red). See *Figure 1—figure supplement 1* and *Figure 1—figure supplement 2A*. (**C**) Similar to (**B**), except that the first extract was supplemented with Alexa568 tubulin (red), the second extract contained no fluorescent tubulin, and the third extract reaction was substituted for purified Cy5 tubulin (magenta) and XMAP215. Branched microtubules (magenta) nucleated from pre-existing microtubules (red), which had been pre-loaded with branching factors in the second extract. See *Figure 1—figure supplement 2B* and *Figure 1—video 2*. For all experiments, images were collected approximately 5 min after the last solution was exchanged. Scale bars, 5 μm. The experiments were repeated three times with different *Xenopus* egg extracts.
The online version of this article includes the following video and figure supplement(s) for figure 1:

**Figure supplement 1.** Testing the inhibitory effect of nocodazole in *Xenopus* egg extract.
**Figure supplement 2.** Sequential *Xenopus* egg extract reactions.
**Figure 1—video 1.** Branching microtubule nucleation from a pre-existing microtubule in *Xenopus* egg extract (related to *Figure 1A*).
https://elifesciences.org/articles/49797#fig1video1
**Figure 1—video 2.** The proteins necessary for branching microtubule nucleation in *Xenopus* egg extract bind to a pre-existing microtubule preceding and independent of the nucleation event (related to *Figure 1C*).
https://elifesciences.org/articles/49797#fig1video2

---

stabilized microtubules and imaged via TIRF microscopy (*Figure 2A*). γ-TuRC, visualized by a fluorescently-labeled antibody, bound along the length of microtubules in the presence of augmin (*Figure 2B–C* and *Figure 2—source data 1*) consistent with previous studies (*Song et al., 2018*). Unexpectedly, TPX2 alone marginally increased the binding of γ-TuRC to microtubules compared to non-specific γ-TuRC binding. A higher amount of γ-TuRC was recruited all along the microtubule lattice in the presence of both TPX2 and augmin (*Figure 2B–C* and *Figure 2—source data 1*). Surprisingly, augmin and TPX2 formed distinct puncta on microtubules, where γ-TuRC was recruited. Using negative stain electron microscopy, we visualized this binding at higher resolution and indeed observed that γ-TuRC is recruited to regularly spaced patches, where it accumulates (*Figure 2D*). Next, we tested whether the microtubule binding proteins augmin and TPX2 need to bind in a certain sequence. Surprisingly, microtubule-bound TPX2 increased the amount of augmin bound to the microtubule, whereas the presence of augmin did not change the level of bound TPX2 (*Figure 2E–F* and *Figure 2—source data 2*). In sum, TPX2 has the unexpected capacity to directly recruit γ-TuRC as well as augmin, which in turn targets more γ-TuRC along the microtubule lattice.

Having established that purified TPX2 and augmin recruit γ-TuRC to template microtubules, can they indeed cause branching microtubule nucleation? All three factors were bound to a stabilized microtubule as above, followed by addition of tubulin and GTP in polymerization buffer (*Figure 3A*). Remarkably, branching microtubule nucleation from a template microtubule occurred using only purified proteins (*Figure 3B* and *Figure 3—video 1*). Live microscopy allowed us to accurately distinguish branching microtubule nucleation from microtubules that were spontaneously nucleated before contacting the microtubule template (*Figure 3—figure supplement 1*). Thus, TPX2, augmin and γ-TuRC are sufficient to specifically nucleate new branched microtubules, which remain attached at the nucleation site on the template microtubule (*Figure 3B*). Occasionally, enough microtubules branched from a single microtubule template (*Figure 3B*, bottom) to create structures reminiscent of those created after TPX2, augmin and γ-TuRC were deposited from *Xenopus* egg extract (*Figure 1—figure supplement 2B*).

How are the nucleation sites spatially organized along the template microtubule? New microtubules nucleate all along the template microtubule without any preference for the template's plus- or minus-ends (*Figure 3C* and *Figure 3—source data 1*), likely because the template microtubule was fully available for simultaneous binding of branching factors in our assay set-up. Thus, there is no signature on the stabilized microtubule lattice that determines where a branch occurs, only that microtubule nucleation events occur from distinct TPX2/augmin puncta distributed along the microtubule lattice (*Figure 3D*). Multiple microtubules can be generated from the same puncta as resolved by light microscopy (*Figure 3D*), which presumably nucleate from neighboring γ-TuRCs (*Figure 2D*). Because both TPX2 and augmin were concurrently tagged with GFP, an unlabeled version of TPX2 was used to confirm that augmin was indeed present in the observed puncta on template microtubules (*Figure 3—figure supplement 2A*).

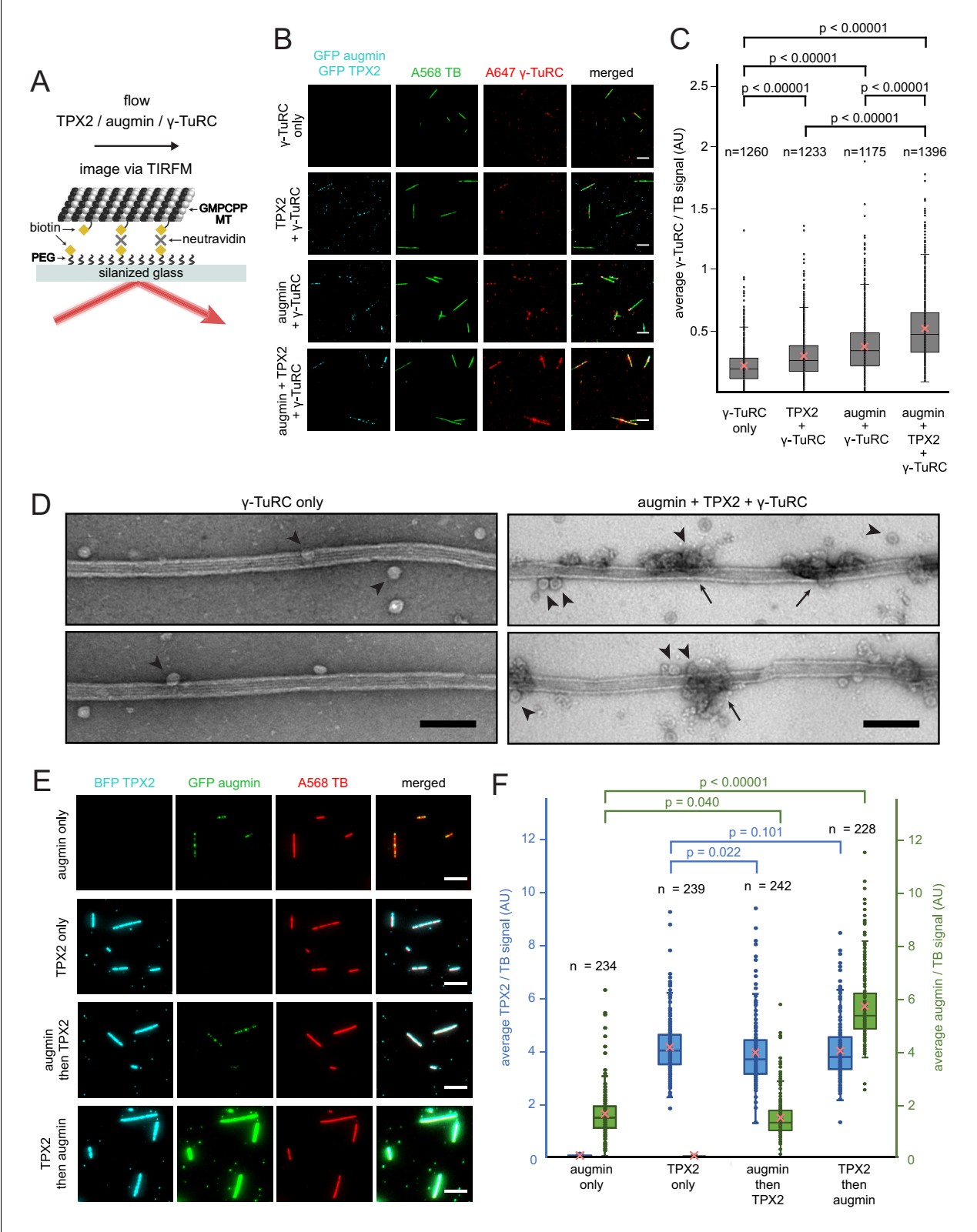

**Figure 2.** Binding of augmin, TPX2 and γ-TuRC to a template microtubule. (**A**) Diagram of the experimental set-up. GMPCPP-stabilized microtubules were attached to a PEG-passivated cover glass with biotin-neutravidin links. (**B**) γ-TuRC (10 nM) visualized using Alexa647-labeled antibodies (red) along microtubules (green), in the absence or presence of GFP-augmin (50 nM) and GFP-TPX2 (50 nM) (cyan). Scale bars, 5 μm. (**C**) Boxplot of average γ-TuRC signal relative to the average tubulin signal, where each dot represents one microtubule from the experiment in (**B**). The number of microtubules (n)

*Figure 2 continued on next page*

*Figure 2 continued*

was obtained from four replicates using γ-TuRC purified from two different preps. (**D**) GMPCPP-stabilized microtubules incubated with γ-TuRC (10 nM) only or with augmin (50 nM), TPX2 (50 nM) and γ-TuRC (10 nM), visualized by electron microscopy after uranyl acetate staining. Ring-shaped structures that correspond to γ-TuRCs (arrowheads), and clusters of protein formed on microtubules (arrows) are visible. Scale bars, 100 nm. (**E**) GFP-augmin (50 nM) (green) and BFP-TPX2 (50 nM) (cyan) visualized along microtubules (red) by themselves or in sequential binding steps. Scale bars, 5 μm. (**F**) Boxplot of average BFP-TPX2 signal or GFP-augmin signal relative to the average tubulin signal, where each dot represents one microtubule from the experiment in (**E**). The number of microtubules (n) was obtained from two replicates. For (**C**) and (**F**), the boxes extend from 25th to 75th percentiles, the whiskers extend from minimum to maximum values, and the mean values are plotted as crosses. P-values were calculated from independent T-tests. The online version of this article includes the following source data and figure supplement(s) for figure 2:

**Source data 1.** Binding of γ-TuRC to a template microtubule.
**Source data 2.** Binding of TPX2 and augmin to a template microtubule.
**Figure supplement 1.** Microtubule nucleation from artificially-attached γ-TuRCs to a template microtubule.
**Figure supplement 2.** Purified TPX2, augmin and γ-TuRC.

What does each protein contribute to branching microtubule nucleation? To test this, each purified factor was assessed alone for its nucleation potential from a template microtubule, combined in pairs and ultimately altogether. Notably, γ-TuRC is essential for branching microtubule nucleation (*Figure 3E* and *Figure 3—source data 2*). Despite the fact that TPX2 can recruit tubulin (*King and Petry, 2019*), it alone or together with augmin cannot nucleate branched microtubules. γ-TuRC can infrequently bind to the microtubule lattice on its own (*Figure 2D*), leading to rare nucleation events without TPX2 and augmin (*Figure 3E* and *Figure 3—source data 2*). Not surprisingly, augmin and γ-TuRC can cause branching microtubule nucleation to a limited extent (*Figure 3E*, *Figure 3—figure supplement 2B–C* and *Figure 3—source data 2*), as augmin can directly recruit γ-TuRC to a pre-existing microtubule in vitro (*Song et al., 2018*). Surprisingly, TPX2 and γ-TuRC can also cause branching microtubule nucleation to a similar extent as augmin and γ-TuRC (*Figure 3E*, *Figure 3—figure supplement 2D–E* and *Figure 3—source data 2*). Although TPX2 only marginally recruits γ-TuRC to the microtubule lattice, its ability to form a co-condensate with tubulin (*King and Petry, 2019*), and to possibly activate γ-TuRC (*Alfaro-Aco et al., 2017*), may facilitate γ-TuRC-dependent microtubule nucleation from a pre-existing microtubule.

Similar to the pattern observed when both TPX2 and augmin are present, augmin or TPX2 alone with γ-TuRC also form distinct puncta along the microtubule lattice, implying that augmin may have a similar mode of binding to the microtubule lattice as TPX2. However, fewer of these puncta actually nucleate branched microtubules (*Figure 3E*, *Figure 3—figure supplement 2C*, *Figure 3—figure supplement 2E* and *Figure 3—source data 2*). Importantly, only when augmin, TPX2 and γ-TuRC are present, branching microtubule nucleation occurs most often (*Figure 3E* and *Figure 3—source data 2*). Branched microtubules are preferentially formed in angles < 90 degrees, with 0–15 degrees being the most common (*Figure 3F* and *Figure 3—source data 3*). This way, most branched microtubules maintain the same polarity as the mother microtubule, a hallmark of branching microtubule nucleation. The angle of microtubule branches did not drastically change when augmin or TPX2 were combined with γ-TuRC, only that augmin/γ-TuRC alone caused a higher proportion of shallow branch angles (*Figure 3F* and *Figure 3—source data 3*).

A key question that we next addressed with this system is whether TPX2 and augmin merely act as localizing factors, or whether they indeed activate γ-TuRC at the branching nucleation sites. To test whether branching was specifically stimulated, we counted the number of non-branched microtubules nucleated in the reactions shown in *Figure 3E*. Reactions where γ-TuRC was present have approximately twice as many non-branched microtubules per field of view compared to the reactions where γ-TuRC was absent. Notably, the addition of TPX2 or augmin does not change the number of non-branched microtubules (*Figure 3—figure supplement 3A* and *Figure 3—source data 4*). This suggests that the increase in branching microtubule nucleation by TPX2 and augmin is not just a result of increased microtubule nucleation overall, but specifically occurs from a pre-existing microtubule.

To more directly assess whether augmin or TPX2 increase the nucleation activity of γ-TuRC, we performed a microtubule nucleation assay in solution. Augmin did not enhance the nucleation activity of purified γ-TuRC in vitro (*Figure 3—figure supplement 3B*), consistent with previous studies (*Song et al., 2018*). A low concentration of TPX2 (10 nM) did not enhance the nucleation activity of

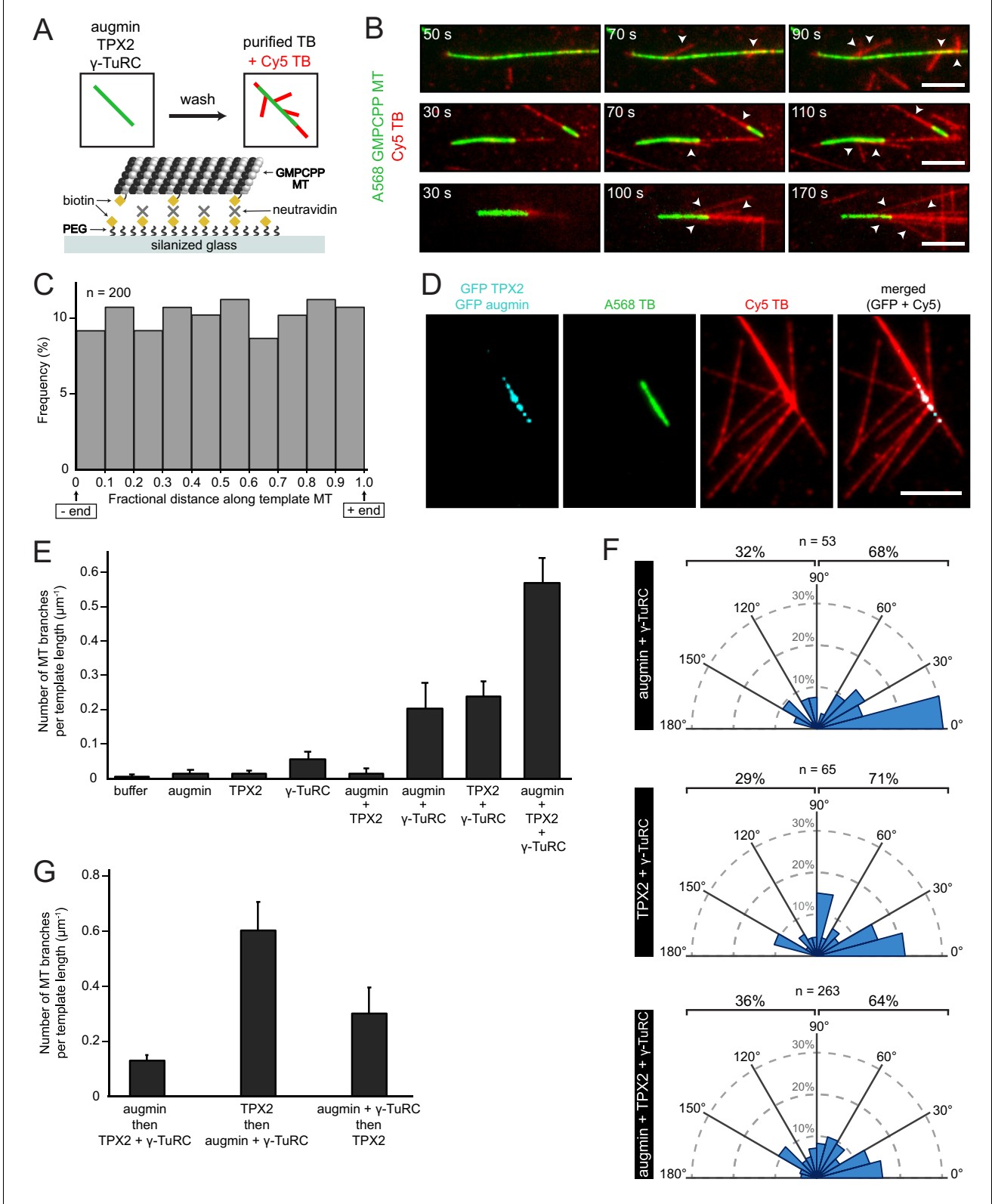

**Figure 3.** Biochemical reconstitution of branching microtubule nucleation using purified augmin, TPX2 and γ-TuRC. (**A**) Diagram of the experimental set-up. GMPCPP-stabilized microtubules were attached to a PEG-passivated cover glass with biotin-neutravidin links. Following the binding of augmin (50 nM), TPX2 (50 nM), and γ-TuRC (10 nM), nucleation of new microtubules was visualized using Cy5 tubulin. (**B**) Using the set-up in (**A**), the formation of microtubule branches (red, arrowheads) from GMPCPP-stabilized microtubules (green) was observed. Scale bars, 5 μm. See *Figure 3—figure*

*Figure 3 continued on next page*

*Figure 3 continued*

*supplement 1A* and *Figure 3—video 1*. (C) Fractional distance along the template microtubule where microtubule branches formed. The 0-point on the x-axis denotes nucleation at the minus-end of the template microtubule, while the 1-point denotes nucleation at the plus-end. The number of branching events (n) was obtained from twelve replicates using γ-TuRC purified from four different preps. (D) Same as (A), microtubule branches (red) grow from distinct GFP-augmin and GFP-TPX2 puncta (cyan) localized on GMPCPP-stabilized microtubules (green). (E) Number of microtubule branches per field of view after 4 min, normalized to the length of template microtubule available, for all the combinations of branching factors. Values are the mean of four replicates using γ-TuRC purified from one prep, and error bars represent standard error of the mean. (F) Angle of branching for three different combinations of branching factors. The number of branching events (n) was obtained from eight replicates using γ-TuRC purified from two different preps in the case of augmin + γ-TuRC and TPX2 + γ-TuRC, and from twelve replicates using γ-TuRC purified from four different preps in the case of augmin + TPX2 + γ-TuRC. (G) Number of microtubule branches per field of view after 4 min, normalized to the length of template microtubule available, for different binding sequences. Values are the mean of four replicates using γ-TuRC purified from one prep, and error bars represent standard error of the mean.

The online version of this article includes the following video, source data, and figure supplement(s) for figure 3:

**Source data 1.** Position of microtubule branches along the template microtubule.
**Source data 2.** Quantification of branched microtubules.
**Source data 3.** Angles of branching microtubule nucleation.
**Source data 4.** Quantification of non-branched microtubules.
**Source data 5.** Quantification of branched microtubules.
**Figure supplement 1.** Microtubules can spontaneously form in solution and subsequently interact with the template GMPCPP-stabilized microtubule.
**Figure supplement 2.** Branching microtubule nucleation with augmin + γ-TuRC and TPX2 + γ-TuRC.
**Figure supplement 3.** Effect of augmin and TPX2 on de novo microtubule nucleation.
**Figure supplement 4.** Reconstitution of branching microtubule nucleation using purified augmin, TPX2, γ-TuRC and XMAP215.
**Figure 3—video 1.** Reconstitution of branching microtubule nucleation using purified augmin, TPX2 and γ-TuRC (related to *Figure 3B*).
https://elifesciences.org/articles/49797#fig3video1

γ-TuRC either (*Figure 3—figure supplement 3C*). A higher concentration of TPX2 (50 nM) was also tested, but at this concentration and in the absence of pre-formed microtubules, TPX2 forms co-condensates with tubulin in solution that nucleate microtubules in vitro (*Figure 3—figure supplement 3D*) (*King and Petry, 2019*), which precludes the accurate assessment of γ-TuRC-mediated microtubule nucleation at higher concentrations. These nucleation experiments suggest that TPX2 and augmin do not activate γ-TuRC in solution. There remains the possibility that they increase the nucleation activity of γ-TuRC specifically when bound to a microtubule. To evaluate whether the increase in branching microtubule nucleation activity in *Figure 3E* is larger than the increase in γ-TuRC binding in *Figure 2C* we calculated the ratio of branching to binding, and normalized it to the γ-TuRC only sample. Although the increase in branching when both TPX2 and augmin are present with γ-TuRC is marginally larger than what is expected from the increase in γ-TuRC localization alone, the difference is not statistically significant based on the current data.

Last, we tested whether branching microtubule nucleation is further enhanced by having XMAP215 present. Indeed, XMAP215 co-localizes to the template microtubule and appears to increase both microtubule nucleation rate and length (*Figure 3—figure supplement 4*). Exact quantification of this effect was not possible because branched microtubules were already formed before imaging was possible and microtubules quickly grew into each other, preventing the accurate identification of branching microtubule nucleation. Lastly, knowing that the binding sequence of TPX2 and augmin matters for maximum factor recruitment, does this have an effect on nucleation? Indeed, only when TPX2 was bound first and augmin/γ-TuRC second, a higher level of branching microtubule nucleation was measured (*Figure 3G* and *Figure 3—source data 5*).

Via an in vitro reconstitution, we demonstrate that the three factors TPX2, augmin and γ-TuRC are sufficient to cause branching microtubule nucleation and defined the roles of each protein. Although microtubule nucleation effectors alone, both as monomers or as phase-separated entities, can generate microtubules in vitro (*Roostalu et al., 2015*; *Woodruff et al., 2017*; *King and Petry, 2019*), γ-TuRC is required for physiological microtubule nucleation (*Kollman et al., 2011*; *Thawani et al., 2018*), and this reconstitution highlights the importance of including it when studying microtubule nucleation. Considered a poor nucleator, γ-TuRC that has been hypothesized to require activation (*Lüders and Stearns, 2007*). In our in vitro system, γ-TuRC's activity could be tuned via the tubulin concentration and the presence of XMAP215. Assay conditions in which not all nucleation events occur at once were necessary in order to clearly identify and observe branching

microtubule nucleation events and to differentiate them from microtubules growing over one another. Based on this, we could investigate the effect of TPX2 and augmin on the activity level of γ-TuRC. The microtubule nucleation capacity of γ-TuRC was not enhanced by TPX2 or augmin in solution, but these factors specifically enable γ-TuRC-dependent microtubule nucleation from a pre-existing microtubule. This can be explained if TPX2 and augmin act as localization factors, but there is also the possibility that they increase the nucleation activity of γ-TuRC specifically when bound to a microtubule. Our results do not conclusively support or rule our either of these models. In the future, single molecule and kinetic measurements along with structural studies are required to determine how exactly TPX2 and augmin interact with γ-TuRC, and whether they increase its activity to stimulate microtubule nucleation from a template microtubule.

Augmin can recruit γ-TuRC to the side of a pre-existing microtubule (*Song et al., 2018*), which can subsequently nucleate branched microtubules. This may be reflective of cell types where TPX2 is not needed for branching microtubule nucleation, such as mitotic *Drosophila* cells (*Verma and Maresca, 2019*). Unexpectedly, TPX2 with γ-TuRC can also cause branching microtubule nucleation to a similar extent as augmin with γ-TuRC. This could be explained by our observation that TPX2 increases γ-TuRC localization to microtubules to a small extent. Additionally, it was recently shown that TPX2 can recruit tubulin to the lattice of a pre-existing microtubule by forming a co-condensate (*King and Petry, 2019*), potentially making lattice-bound γ-TuRC more likely to nucleate branched microtubules. Finally, TPX2 contains γ-TuRC activator motifs that have been proposed to activate γ-TuRC (*Alfaro-Aco et al., 2017*). All three of these functions may be the reason why TPX2, which recruited less γ-TuRC than augmin, subsequently caused γ-TuRC-dependent branching microtubule nucleation at a similar level as augmin. Besides operating via these three modes in the full reconstitution, TPX2 additionally recruits more augmin, which in turn can localize more γ-TuRC. Finally, TPX2 alone creates a slightly wider angle distribution of branches, than augmin alone and both factors together. Thus, by bridging the microtubule and γ-TuRC with a defined architecture based on its h-shaped structure (*Hsia et al., 2014*; *Song et al., 2018*), augmin may keep the angle variation in a more shallow range, which is critical for creating parallel structures that retain microtubule polarity as is necessary in the spindle and axons.

Although TPX2 + γ-TuRC or augmin + γ-TuRC can nucleate microtubules from a pre-existing microtubule in this purified system, both factors are required in Xenopus egg extract for branching microtubule nucleation (*Petry et al., 2013*). This indicates that microtubule nucleation in a physiological environment is more stringently regulated. Another important difference is that while it is beneficial that TPX2 binds to microtubules before augmin in our in vitro reconstitution, TPX2 binding to microtubules is absolutely necessary for augmin/γ-TuRC binding and subsequent branching microtubule nucleation in Xenopus egg extract (*Thawani et al., 2019*). This implies that TPX2 directly regulates augmin's binding to microtubules in Xenopus egg extract, in a manner that still needs to be uncovered.

In summary, TPX2, augmin and γ-TuRC are required for efficient branching microtubule nucleation in vitro. Surprisingly, TPX2, despite being only a single protein amongst large multi-subunit complexes, lies at the very heart of controlling this reaction, as demonstrated in this purified in vitro system and in Xenopus egg extract (*Thawani et al., 2019*). Finally, even though TPX2, augmin and γ-TuRC are sufficient to reconstitute the nucleation of branched microtubules in vitro, we cannot eliminate the possibility that other proteins are involved in additional regulation, such as the microtubule and augmin binding protein EML3 (*Luo et al., 2019*).

Mechanistic insight obtained from this work helps explain how microtubules are made within spindles to orchestrate chromosome segregation. It is analogous to the in vitro reconstitution of actin branching (*Mullins et al., 1998*), which paved the way to explain how the actin cytoskeleton supports cell locomotion. In this biochemical reconstitution, γ-TuRC was localized to a specific location, from which it nucleates a microtubule in vitro as it would occur in the cell. As such, this work serves as a platform to study how microtubule nucleation creates different microtubule architectures that support cell function. This approach presents a pioneering example for the biochemical reconstitution of other microtubule nucleation pathways or microtubule organizing centers. It also serves as a stepping-stone to reconstitute larger structures based on this microtubule nucleation pathway, such as the mitotic spindle.

## Materials and methods

### Cloning, expression and purification of proteins

DH5α *E. coli* cells (New England Biolabs, C2987I) were used for all cloning steps. Rosseta2 (DE3) pLysS cells (Novagen, 714034) were used for all protein expression in *E. coli*, and cultures were grown in TB Broth (Sigma-Aldrich, T0918), or in LB Broth (Sigma-Aldrich, L3522) for the expression of TPX2. Sf9 cells using the Bac-to-Bac system (Invitrogen) were used in the expression of augmin and XMAP215, and cultures were grown in Sf-900 III SFM (Gibco, 12658027).

Human RanQ69L with N-terminal Strep-6xHis-BFP, and human EB1 with C-terminal GFP-6xHis were expressed and purified as previously described (*Thawani et al., 2019*). Full-length *Xenopus laevis* TPX2 constructs were expressed and purified as previously described (*King and Petry, 2019*). Briefly, N-terminal Strep-6xHis-GFP TPX2 and Strep-6xHis-BFP TPX2 were cloned into pST50 vectors and expressed in *E. coli* for 7 hr at 25 °C. Both proteins were affinity purified using Ni-NTA agarose beads (Qiagen, 30250) followed by gel filtration with a Superdex 200 HiLoad 16/600 column (GE Healthcare) in CSF-XB buffer (100 mM KCl, 10 mM K-HEPES, 1 mM $MgCl_2$, 0.1 mM $CaCl_2$, 5 mM EGTA, pH 7.7) + 10% w/v sucrose. Unlabeled TPX2 was generated by cleaving Strep-6xHis-GFP with TEV protease at 100:1 TPX2:TEV protease molar ratio overnight at 4 °C. Full-length *Xenopus laevis* XMAP215 with C-terminal GFP-7xHis was expressed in Sf9 cells using the Bac-to-Bac system and purified as previously described (*Thawani et al., 2018*). Briefly, XMAP215 was affinity-purified using a HisTrap HP 5 ml column (GE Healthcare), followed by cation-exchange with a Mono S 10/100 GL column (GE Healthcare). The protein was dialyzed overnight into CSF-XB + 10% w/v sucrose. GFP-tagged *Xenopus laevis* augmin holocomplex was co-expressed in Sf9 cells using the Bac-to-Bac system and purified as previously described (*Song et al., 2018*). Briefly 1–2 liters of Sf9 cells (1.5–2.0 × $10^6$ $mL^{-1}$) were co-infected with different baculoviruses, each carrying a subunit of the augmin complex, at MOIs of 1–3. Cells were collected 72 hr after infection. HAUS6 had an N-terminal ZZ-tag and HAUS2 had a C-terminal GFP-6xHis. The remaining six subunits were untagged. Augmin holocomplex was affinity-purified using IgG-Sepharose (GE Healthcare, 17-0969-01) and eluted via cleavage with 100–200 μg of GST-HRV3C protease. The HRV3C protease was subsequently removed using a GSTrap 5 mL column (GE Healthcare). The sample was further purified and concentrated using Ni-NTA agarose beads. The protein was dialyzed overnight into CSF-XB + 10% w/v sucrose. All recombinant proteins were flash frozen and stored at −80 °C. Protein concentrations were determined with Bradford dye (Bio-Rad, 5000205) or using a Coomassie-stained SDS-PAGE gel loaded with known concentrations of BSA (Sigma-Aldrich, B6917).

Native γ-TuRC was purified from Xenopus egg extract with some changes to previously described protocols (*Zheng et al., 1995*; *Thawani et al., 2018*). 5 ml of Xenopus egg extract were diluted 10-fold with CSF-XB + 10% w/v sucrose, 1 mM GTP, 1 mM DTT, and 10 μg $ml^{-1}$ leupeptin, pepstatin and chymostatin. Large particles were removed by spinning at 3000 *g* for 10 min at 4°C. The supernatant was further diluted two-fold with buffer and passed through filters of decreasing pore size (1.2 μm, 0.8 μm and 0.22 μm). γ-TuRC was precipitated from the filtered extract by addition of 6.5% w/v polyethylene glycol (PEG) 8000 and incubated on ice for 30 min. After centrifugation for 20 min at 17,000 *g* at 4°C, the pellet was resuspended in 15 ml of the initial CSF-XB buffer supplemented with 0.05% NP-40. The resuspended pellet was centrifuged at 136,000 *g* at 4°C for 7 min. The supernatant was then precleared with protein A Sepharose beads (GE Healthcare, 45002971) for 20 min at 4°C. The beads were removed by spinning, 2–4 ml γ-tubulin antibody (1 mg $ml^{-1}$) was added to the sample, and the sample was rotated at 4°C for 2 hr. After this, 1 ml of washed Protein A Sepharose beads was incubated with the sample on the rotator for 2 hr at 4°C. The beads were collected by spinning, and subsequently transferred to a column with the same buffer used to resuspend the PEG pellet. The beads were washed with the initial CSF-XB buffer supplemented with extra 150 mM KCl, then with CSF-XB buffer supplemented with 1 mM ATP, and finally with CSF-XB buffer to remove the ATP. For biotinylation of γ-TuRC, the beads were incubated with 25 μM of NHS-PEG4-biotin (Thermo Scientific, A39259) in CSF-XB buffer for 1 hr at 4°C, and unreacted reagent was washed away with CSF-XB buffer before elution with γ-tubulin peptide. 2 ml γ-tubulin peptide (amino acids 412–451) at 0.5 mg $ml^{-1}$ in CSF-XB buffer was applied to the column and allowed to incubate overnight. The eluted sample was collected the following day, and it was concentrated using a 100 kDa MWCO centrifugal-filter (Amicon, UFC810024). This concentrated sample was loaded onto a 10–50% w/w sucrose gradient in the initial CSF-XB buffer, and centrifuged at 200,000

*g* for 3 hr at 4°C in a TLS55 rotor (Beckman Coulter). The sucrose gradient was fractionated manually from the top, and the fractions with the highest γ-tubulin signal by Western blotting were combined and concentrated using another 100 kDa MWCO centrifugal-filter. Purified γ-TuRC was always used within two days on ice without freezing.

Unlabeled cycled tubulin purified from bovine brain was obtained from a commercial source (Pur-Solutions, 032005). Before use, all proteins were pre-cleared of aggregates via centrifugation at 80,000 RPM for 15 min at 4 °C in a TLA100 rotor (Beckman Coulter).

## Tubulin labeling and polymerization of GMPCPP-stabilized microtubules

Bovine brain tubulin was labeled following previously described methods (*Hyman et al., 1991*). Using Cy5-NHS ester (GE Healthcare, PA15101) yielded 54–70% labeling. Using Alexa-568 NHS ester (Invitrogen, A20003) yielded 36–40% labeling. Labeling efficiency with biotin-PEG4-NHS (Thermo Scientific, A39259) was not calculated.

Single-cycled GMPCPP-stabilized microtubules were made as previously described (*Gell et al., 2010*). Briefly, 12 µM unlabeled tubulin + 1 µM Alexa-568 tubulin + 1 µM biotin tubulin was polymerized in BRB80 (80 mM Pipes, 1 mM EGTA, 1 mM MgCl$_2$) in the presence of 1 mM GMPCPP (Jena Bioscience, NU-405L) for 1 hr at 37°C. For GMPCPP-stabilized microtubules without any labels, 14 µM unlabeled tubulin was polymerized. For GMPCPP-stabilized microtubules without biotin, 13 µM unlabeled tubulin + 1 µM Alexa-568 tubulin was polymerized.

## Preparation of polyethylene glycol (PEG)-functionalized surfaces

Cover glasses (Carl Zeiss, 474030-9020-000) were silanized and reacted with PEG as previously described (*Bieling et al., 2010*), except that hydroxyl-PEG-3000-amine (Rapp Polymere, 103000–20) and biotin-PEG-3000-amine (Rapp Polymere, 133000-25-20) were used. Glass slides were passivated with poly(L-lysine)-PEG (SuSoS) (*Bieling et al., 2010*). Flow chambers for TIRF microscopy were assembled using double-sided tape.

## Attachment of GMPCPP-stabilized microtubules to PEG-functionalized surfaces

The assay was performed following a previously described protocol with some changes (*Roostalu et al., 2015*). Flow chambers were incubated with 5% Pluronic F-127 in water (Invitrogen, P6866) for 10 min at room temperature and then washed with assay buffer (80 mM Pipes, 30 mM KCl, 1 mM EGTA, 1 mM MgCl$_2$, 1 mM GTP, 5 mM 2-mercaptoethanol, 0.075% (w/v) methylcellulose (4000 cP; Sigma-Aldrich, M0512), 1% (w/v) glucose, 0.02% (v/v) Brij-35 (Thermo Scientific, 20150)) supplemented with 50 µg mL$^{-1}$ k-casein (Sigma-Aldrich, C0406) and extra 0.012% (v/v) Brij-35. Flow chambers were then incubated with assay buffer containing 50 µg mL$^{-1}$ NeutrAvidin (Invitrogen, A2666) for 3 min on a metal block on ice and subsequently washed with BRB80 (80 mM Pipes, 1 mM EGTA, 1 mM MgCl$_2$). Next, flow chambers were incubated for 5 min at room temperature with bio-tin- and Alexa-568-labeled GMPCPP-stabilized microtubules diluted 1:2000 in BRB80. Unbound microtubules were removed by subsequent BRB80 washes.

## Binding of proteins to GMPCPP-stabilized microtubules

To test the recruitment of γ-TuRC to a microtubule by augmin and TPX2, a mixture of GFP-TPX2 (50 nM), GFP-augmin (50 nM) and γ-TuRC (10 nM), which was previously incubated for 5 min on ice, was added to a flow chamber that had GMPCPP-stabilized microtubules attached to the surface as described above. This was incubated for 5 min at room temperature. Unbound proteins were removed with additional BRB80 washes. To visualize native γ-TuRC, Alexa-647 (Invitrogen) labeled antibodies against γ-tubulin (XenC antibody, 2 µg ml$^{-1}$) were added to the flow chamber and incubated for 10 min at room temperature. Unbound antibody was removed with additional BRB80 washes, and the final solution was exchanged to BRB80 + 250 nM glucose oxidase (Crescent Chemical, SE22778.02), 64 nM catalase (Sigma-Aldrich, C40) and 1% (w/v) glucose. The sample was imaged immediately. For experiments where one or two of the proteins in the mixture were omitted, the volume was substituted with CSF-XB buffer. The same set-up was used when imaging the binding of GFP-augmin (50 nM) and GFP-TPX2 (50 nM) to microtubules in the presence of each other. In these cases, γ-TuRC was not included, and instead of adding XenC antibody, BRB80 + oxygen

scavengers were added after the last unbound proteins were removed by BRB80 washes. In experiments where two proteins were bound to microtubules sequentially, unbound protein was removed by BRB80 washes before the second protein was added.

## Microtubule nucleation assays on PEG-functionalized surfaces

For branching microtubule nucleation reactions in vitro, a mixture of TPX2 (50 nM), augmin (50 nM) and γ-TuRC (10 nM), which was previously incubated for 5 min on ice, was added to the chamber containing attached GMPCPP-stabilized microtubules and incubated for 5 min at room temperature. Unbound proteins were removed by additional BRB80 washes. The final assay mixture was flowed into the chambers: 80 mM Pipes, 30 mM KCl, 1 mM EGTA, 1 mM MgCl$_2$, 1 mM GTP, 5 mM 2-mercaptoethanol, 0.075% (w/v) methylcellulose (4000 cP), 1% (w/v) glucose, 0.02% (v/v) Brij-35, 250 nM glucose oxidase, 64 nM catalase, 1 mg ml$^{-1}$ BSA, 19 μM unlabeled bovine tubulin and 1 μM Cy5-labeled bovine tubulin. For experiments where XMAP215-GFP was added to this final reaction its concentration was 50 nM.

## Microtubule nucleation from artificially-attached γ-TuRCs to microtubules

Coverslips were coated with dichlorodimethylsilane (*Gell et al., 2010*). Flow chambers for TIRF microscopy were assembled using double-sided tape and incubated for 5 min at room temperature with biotin- and Alexa-568-labeled GMPCPP-stabilized microtubules diluted 1:2000 in BRB80. A small number of microtubules attached non-specifically to the glass, and the rest were removed with BRB80 washes. The rest of the glass surface was blocked with 1% Pluronic F127, and the chamber was incubated for 3 min at room temperature with 500 μg mL$^{-1}$ NeutrAvidin diluted in BRB80. Undiluted biotinylated γ-TuRC (10 nM) was incubated in the chamber for 10 min at room temperature, and after washing with BRB80 the final tubulin nucleation mix was added: 80 mM Pipes, 1 mM EGTA, 1 mM MgCl$_2$, 1 mM GTP, 2.5 mM PCA, 25 nM PCD, 2 mM Trolox, 19 μM unlabeled bovine tubulin and 1 μM Cy5-labeled bovine tubulin.

## Microtubule nucleation assays in solution

Microtubule nucleation reactions containing 9.5 μM unlabeled bovine tubulin, 0.5 μM Cy5-labeled bovine tubulin and 1.5 mM GTP in BRB80, were mixed with augmin, TPX2 or γ-TuRC. CSF-XB buffer was used when proteins were not added. Augmin was added at 50 nM, TPX2 was added at 10 nM or 50 nM, and γ-TuRC was added at 5 nM. The reactions were incubated at 37˚C for 5 min. The nucleation mixture was then diluted 5-fold with warm BRB80, immediately diluted again 2-fold with 2% glutaraldehyde in BRB80, and left at RT for 5 min. The samples were then diluted 5-fold in BRB80, layered on top of a 5 ml cushion of 20% (vol/vol) glycerol in BRB80 prepared in a 15 ml Corex tube fitted with a custom insert to support a round poly-lysine–treated coverslip. The sample was centrifuged for 45 min at 25,000 g in an HB-6 rotor at 4˚C. After centrifugation, the cushion was removed, and ice-cold methanol was added to the tube. The coverslip was mounted in Prolong Diamond (Invitrogen, P36965).

## Sequential xenopus egg extract reactions

CSF extracts were prepared from *Xenopus laevis* oocytes as described previously (*Murray and Kirschner, 1989*; *Hannak and Heald, 2006*). When working with *Xenopus laevis*, all relevant ethical regulations were followed, and all procedures were approved by Princeton IACUC. Extract reactions were done in flow chambers prepared between glass slides and 22 × 22 mm, 1.5 coverslips (Fisherbrand, 12-541B) using double-sided tape. In all reactions 75% of the total volume was extract, and 25% was a combination of other components or CSF-XB (100 mM KCl, 10 mM K-HEPES, 1 mM MgCl$_2$, 0.1 mM CaCl$_2$, 5 mM EGTA, pH 7.7) + 10% w/v sucrose. All reactions were done in the presence of 0.5 mM sodium orthovanadate (NEB, P0758S) to avoid sliding of microtubules on the glass surface, and with 0.89 μM fluorescently-labeled tubulin. In reactions where BFP-RanQ69L was added, its concentration was 10 μM. When EB1-GFP was added its concentration was 85 nM. All proteins and chemicals added to egg extracts were stored or diluted into CSF-XB buffer + 10% w/v sucrose. Reaction mixtures were pipetted into the flow chambers to initiate microtubule formation.

For sequential extract reactions, individual microtubules were allowed to form on the glass surface from the first extract reaction for 5–8 min, and soluble, non-microtubule bound proteins were removed by washing with CSF-XB. For experiments with three sequential reactions, the CSF-XB wash was supplemented with 0.05 mM Taxol (Sigma-Aldrich, T7402). The second extract reaction was then introduced. In the case of *Figure 1a* the chamber was imaged immediately. For all other experiments, the second extract with 0.033 mM nocodazole (Sigma-Aldrich, M1404) was incubated in the chamber for 5 min, followed by the removal of unbound protein with CSF-XB if the third reaction was extract, or with BRB80 if the third reaction was purified tubulin. The third extract reaction was then introduced and imaged immediately. For experiments where the final reaction was purified tubulin, the final tubulin nucleation mix was added: 80 mM Pipes, 1 mM EGTA, 1 mM MgCl$_2$, 1 mM GTP, 2.5 mM PCA, 25 nM PCD, 2 mM Trolox, 19 µM unlabeled bovine tubulin and 1 µM Cy5-labeled bovine tubulin. If XMAP215-GFP was added in this final reaction, its concentration was 25 nM.

## TIRF microscopy and image analysis

Total internal reflection fluorescence (TIRF) microscopy was performed with a Nikon TiE microscope using a 100 × 1.49 NA objective. Andor Zyla sCMOS camera was used for acquisition, with a field of view of 165.1 × 139.3 µm. 2 × 2 binned, multi-color images were acquired using NIS-Elements software (Nikon). All adjustable imaging parameters (exposure time, laser intensity, and TIRF angle) were kept the same within experiments. For microtubule nucleation assays in vitro the TIRF objective was warmed to 33°C using an objective heater (Bioptechs, 150819–13). For all time-lapse imaging, multi-color images were collected every 2 s. Brightness and contrast were optimized individually for display, except for images in *Figure 1—figure supplement 1*, *Figure 2*, *Figure 3—figure supplement 3* and *Figure 3—figure supplement 4*, where images belonging to the same experiment were contrast-matched.

Images used for the quantification of microtubule binding were analyzed using ImageJ (*Schindelin et al., 2012*). To segment microtubules, the tubulin signal was first thresholded via the Otsu method. Microtubules were isolated from the mask by setting the minimum particle area as 1 or 2 µm$^2$. Average fluorescent signals per pixel, for the microtubule or bound proteins, were calculated for each microtubule. The average intensity from the reverse mask of the entire field of view was subtracted from the average intensity on each microtubule. For branching microtubule nucleation experiments in vitro, microtubules were counted manually using time-lapse experiments within the first 3.5 min of the reaction. Lengths of microtubules and branching angles were measured using ImageJ.

## Negative stain electron microscopy

Unlabeled GMPCPP-stabilized microtubules diluted 1:500 were incubated for 5 min at room temperature with either γ-TuRC only or with a mixture of TPX2 (50 nM) + augmin (50 nM) + γ-TuRC. The samples were diluted 10-fold with BRB80 to reduce the number of unbound γ-TuRC molecules in the background, and 5 µl of this diluted sample was immediately applied onto glow-discharged grids (Electron Microscopy Sciences, CF400-Cu). The samples were stained with 2% uranyl acetate. Images were collected with a CM100 TEM (Philips) at 80 keV at a magnification of 64,000. Images were recorded using an ORCA camera.

## Antibodies

Polyclonal XenC antibody was a gift from C. Wiese and was described previously (*Wiese and Zheng, 2000*). It was used to generate Alexa-647-labeled XenC antibody by first dialyzing antibodies in PBS buffer (50 mM NaPO$_4$, 150 mM NaCl, pH 7.4). The reaction with Alexa-647-NHS-ester was done according to the protocol recommended by the manufacturer. Finally, the removal of unreacted dye was done via gel filtration in Bio-Gel P-30 Gel (Bio-Rad). On average, each XenC antibody was labeled with 2.5 Alexa-647 dye molecules. The polyclonal antibody used to purify γ-TuRC from Xenopus egg extract was generated against a purified γ-tubulin peptide (amino acids 412–451) through a commercial vendor (Genscript). The presence of γ-TuRC during its purification was tracked via Western blotting using the GTU88 (Sigma-Aldrich, T6557) antibody against γ-tubulin.

## Acknowledgements

We are grateful to Christiane Wiese for providing XenC antibodies, Jae-Geun Song and Brian Mahon for help in the expression and purification of augmin, Matt King for help in the expression and purification of TPX2, all members of the Petry laboratory for discussions, Thomas Surrey for sharing the detailed protocol for making biotin-PEG-functionalized coverslips, and James Wakefield for sharing unpublished data, critically reading this manuscript and for coordinating submissions of manuscripts to eLife. We thank Ron Vale with whom the original project vision was conceived. This work was supported by the HHMI Gilliam Fellowship and the NSF Graduate Research Fellowship (to RA), the American Heart Association predoctoral fellowship 17PRE33660328 (to AT), the NIH New Innovator Award, the Pew Scholars Program in the Biomedical Sciences, and the David and Lucile Packard Foundation (to SP).

## Additional information

### Funding

| Funder | Grant reference number | Author |
| --- | --- | --- |
| National Institute of General Medical Sciences | T32GM007388 | Raymundo Alfaro-Aco |
| Howard Hughes Medical Institute | Gilliam Fellowship | Raymundo Alfaro-Aco |
| National Science Foundation | Graduate Research Fellowship | Raymundo Alfaro-Aco |
| American Heart Association | 17PRE33660328 | Akanksha Thawani |
| National Institute of General Medical Sciences | 1DP2GM123493-01 | Sabine Petry |
| David and Lucile Packard Foundation | 2014-40376 | Sabine Petry |

The funders had no role in study design, data collection and interpretation, or the decision to submit the work for publication.

### Author ORCIDs

Raymundo Alfaro-Aco https://orcid.org/0000-0003-4196-3797
Akanksha Thawani https://orcid.org/0000-0003-4168-128X
Sabine Petry https://orcid.org/0000-0002-8537-9763

### Ethics

Animal experimentation: This study was performed in strict accordance with the recommendations in the Guide for the Care and Use of Laboratory Animals of the National Institutes of Health. All of the animals were handled according to approved Institutional Animal Care and Use Committee (IACUC) protocol # 1941-16 of Princeton University.

### Author contributions

Raymundo Alfaro-Aco, Conceptualization, Resources, Data curation, Formal analysis, Validation, Investigation, Visualization, Methodology, Writing - original draft, Writing – review and editing; Akanksha Thawani, Resources, Methodology, Writing—review and editing; Sabine Petry, Conceptualization, Supervision, Funding acquisition, Writing—original draft, Project administration, Writing—review and editing

### Decision letter and Author response

Decision letter https://doi.org/10.7554/eLife.49797.sa1
Author response https://doi.org/10.7554/eLife.49797.sa2

## Additional files

### Supplementary files
• Transparent reporting form

### Data availability
All data generated or analyzed during this study are included in the manuscript and supporting files. Source data files have been provided for Figures 2 and 3.

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
