## [Decision Letter]

**Acceptance summary:**

In this manuscript, the authors use *Xenopus* augmin recombinantly produced in insect cells, purified native γ-TuRC from *Xenopus* egg extract, and recombinant TPX2 purified from *E. coli* to reconstitute the nucleation of microtubules from the lateral surface of pre-existing microtubules. This process, known as branching microtubule nucleation, is important, for example, for spindle formation and neuronal development. The authors show the same requirements for TPX2, augmin and γ-TuRC that was seen previously in extracts and suggest that these components are sufficient for branching nucleation. The work thus represents a useful step forward in our understanding of multicomponent microtubule nucleation and will be of interest to a broad community of cell biologists and biochemists.

**Decision letter after peer review:**

Thank you for submitting your article "Biochemical reconstitution of branching microtubule nucleation" for consideration by *eLife*. Your article has been reviewed by three peer reviewers, including Jens Lüders as the Reviewing Editor and Reviewer #1, and the evaluation has been overseen by Anna Akhmanova as the Senior Editor.

The reviewers have discussed the reviews with one another and the Reviewing Editor has drafted this decision to help you prepare a revised submission.

Summary:

The manuscript "Biochemical reconstitution of branching microtubule nucleation" by Alfaro-Aco et al. uses *Xenopus* augmin recombinantly produced in insect cells, purified native γ-TuRC from *Xenopus* egg extract, and recombinant TPX2 purified from *E. coli* to reconstitute the nucleation of microtubules from the lateral surface of pre-existing microtubules.

Since the discovery of branching microtubule nucleation and its key player augmin, this nucleation mechanism has been analyzed largely in cell biological studies and a true mechanistic understanding is still lacking.

A set of papers from the Petry lab culminating with this manuscript has explored the contributions of augmin, TPX2 and γ-TuRC to branching microtubule nucleation. This manuscript shows the same requirement for TPX2, augmin and γ-TuRC that was seen previously in extracts and suggests that these components are sufficient for branching nucleation.

Essential revisions:

Overall, this is a well written, timely manuscript that establishes the minimal molecular requirements for branching microtubule nucleation. The reconstitution of the process by purified proteins constitutes an important achievement in the field and provides the basis for further mechanistic investigation.

However, the reviewers felt that the study lacks some novelty considering the authors' recently published reconstitution of augmin-dependent recruitment of γ-TuRC using the same purified components (Song et al., 2018). To address this the efficiency of the reconstitution of branching nucleation should be determined. A related issue is the question if and how γ-TuRC is activated in the reconstituted system and what the roles of TPX2 and augmin are in this regard.

1) Validating the success of reconstitution requires knowing the efficiency of the reconstitution. If only a small fraction of the proteins is active, then the reconstitution may be incomplete. Concentrations need to be reported for γ-TuRC with an estimate of the fraction of active γ-TuRCs.

2) Do TPX2 and augmin merely function as recruitment factors or do they activate γ-TuRC at the branching nucleation sites? Figure 3 - video 1: It is evident that in the optimal branching nucleation mix with TPX2, augmin, and γ-TuRC, a large number of microtubules are nucleated spontaneously and not from the lattice of existing microtubules. Unfortunately, only branching events are quantified in Figure 3E. Is branching specifically stimulated or is there an overall increased nucleation activity, which may also explain more branches? γ-TuRC alone is generally considered a weak nucleator, so the obvious question is what activates γ-TuRC in these assays? This could also be tested in a regular nucleation assay (without preformed microtubules).

3) It is hard to believe that TPX2 and γ-TuRC, without augmin, can induce branching nucleation, given that TPX2 has no ability to recruit γ-TuRC to existing microtubules (Figure 2—figure supplement 2). The authors show that the branch angle is slightly different from augmin-dependent nucleation (Figure 3F). Is this the only difference? The authors should investigate this peculiar phenomenon more thoroughly.

[Editors' note: further revisions were requested prior to acceptance, as described below.]

Thank you for resubmitting your work entitled "Biochemical reconstitution of branching microtubule nucleation" for further consideration by *eLife*. Your revised article has been evaluated by Anna Akhmanova (Senior Editor) and a Reviewing Editor.

All reviewers agree that the revisions have much improved the manuscript, but there are two remaining issues that need to be addressed before acceptance.

In the manuscript you argue that TPX2 and augmin are activation factors for γ-TuRC and not just localization factors. The evidence is that TPX2 and augmin do not activate γ-TuRC based nucleation away from the MT lattice, but do increase nucleation of branches from the MT lattice. However, this would actually be the definition of a localization factor, to give more nucleation from a specific location, without requiring activation. When comparing Figure 2C and 3E it seems that there is a stronger increase in branches (Figure 3E) than there is an increase in γ-TuRC on the lattice (Figure 2C). Given the size of the error and the spread of the data, is this a significant difference? At the very least it seems to be a small activation. We would like you to address this point more clearly in the paper. Also, if your conclusion is that the activation is indeed very small, you may want to consider changing the Abstract sentence "TPX2 and augmin do not act as mere localization factors, but enable γ-TuRC-dependent microtubule nucleation at preferred branching angles of less than 90 degrees from regularly-spaced patches along microtubules." to "TPX2 and augmin enable γ-TuRC-dependent microtubule nucleation at preferred branching angles of less than 90 degrees from regularly-spaced patches along microtubules."

We would also like to ask you to include the data shown in your rebuttal letter as part of Figure 3—figure supplement 2 (GFP-augmin, untagged TPX2 and γ-TuRC).

---

## [Author Response]

Essential revisions:Overall, this is a well written, timely manuscript that establishes the minimal molecular requirements for branching microtubule nucleation. The reconstitution of the process by purified proteins constitutes an important achievement in the field and provides the basis for further mechanistic investigation.However, the reviewers felt that the study lacks some novelty considering the authors' recently published reconstitution of augmin-dependent recruitment of γ-TuRC using the same purified components (Song et al., 2018). To address this the efficiency of the reconstitution of branching nucleation should be determined. A related issue is the question if and how γ-TuRC is activated in the reconstituted system and what the roles of TPX2 and augmin are in this regard.

We thank the Reviewing Editor for this summary and for the excellent and helpful review process. We addressed all of these comments and thereby generated an improved manuscript, as explained in more detail below. The new manuscript provides more insight into the mechanism of branching microtubule nucleation by addressing the activity level of γ-TuRC and the role of TPX2 and augmin in enabling microtubule nucleation from a pre-existing microtubule.

1) Validating the success of reconstitution requires knowing the efficiency of the reconstitution. If only a small fraction of the proteins is active, then the reconstitution may be incomplete. Concentrations need to be reported for γ-TuRC with an estimate of the fraction of active γ-TuRCs.

We thank the reviewers for this excellent comment. We fully agree that this is an important issue that needs to be assessed and discussed.

First, the concentration of γ-TuRC added in each experiment along with the concentrations of the other proteins is now reported in each figure legend. In addition, a supplementary figure (Figure 2—figure supplement 2) was added, which displays SDS PAGE gels of the purified complexes and proteins, as well as a negative stain EM image of purified γ-TuRC that highlights the quality of the purified complexes.

Estimating the fraction of active γ-TuRC is a complex and challenging question that can be addressed at multiple levels.

i) The higher the tubulin concentration, the more γ-TuRCs will fire. However, when more γ-TuRCs fire, the smaller the time and “field of view” windows to observe that microtubules indeed branch as opposed to grow into/over a microtubule.

ii) Analogous to point (i), addition of XMAP215 visually increases the number of microtubule branches, suggesting that at least some of the γ-TuRCs that are not nucleating microtubules are in fact capable of doing so under more favourable conditions. However, this again makes the observation of true branches in space and time extremely difficult.

iii) In the negative stain EM images from Figure 2, we can see multiple γ-TuRCs recruited along the microtubule lattice and often to the same puncta on pre-existing microtubules. If we crudely compare the number of γ-TuRCs from these EM images with the number of microtubule branches in Figure 3, it is clear that most γ-TuRCs are not nucleating microtubules in our conditions. Again, these conditions are necessary to clearly observe branching microtubule nucleation take place and to differentiate it from other events.

iv) Last, because our native γ-TuRC purification cannot easily be equipped with a fluorescent tag, it is not possible to differentiate active versus non-active γ-TuRC using light microscopy. Even if that was possible, one could not exclude that the labeled γ-TuRC that did not fire is indeed fully intact and contains all subunits.

All of these points suggest that not all γ-TuRCs nucleate in these conditions. In fact, tuning the conditions so that microtubule nucleation by γ-TuRC, as well as spontaneous nucleation, is not happening all at once, e.g. by omitting XMAP215 and working at moderate tubulin concentrations, was necessary to make the findings reported in this manuscript. This is now clearly addressed both in the Results and in the Discussion section of the manuscript. We recognize that this comment is forming the basis of the next point (2), about whether TPX2 or augmin activate γ-TuRC, which we reply to below. Finally, it is possible that other proteins may be involved in increasing the efficiency of branching microtubule nucleation. We also better addressed this possibility in the discussion of the improved manuscript.

2) Do TPX2 and augmin merely function as recruitment factors or do they activate γ-TuRC at the branching nucleation sites? Figure 3 - video 1: It is evident that in the optimal branching nucleation mix with TPX2, augmin, and γ-TuRC, a large number of microtubules are nucleated spontaneously and not from the lattice of existing microtubules. Unfortunately, only branching events are quantified in Figure 3E. Is branching specifically stimulated or is there an overall increased nucleation activity, which may also explain more branches? γ-TuRC alone is generally considered a weak nucleator, so the obvious question is what activates γ-TuRC in these assays? This could also be tested in a regular nucleation assay (without preformed microtubules).

We thank the reviewers for highlighting these important questions. First, it is true that nonbranched microtubules form in the experiments described in Figure 3. Some of these microtubules are spontaneously generated in solution, as they grow from both ends and diffuse around. Others are nucleated by γ-TuRCs that are non-specifically bound to the glass surface, as these microtubules would not form without γ-TuRC, and they remain attached at the site of nucleation where only one end grows. We now quantified these non-branched microtubules and the graph is shown in a new Figure 3—figure supplement 3. The reactions that contain γ-TuRC have approximately twice as many non-branched microtubules, compared to the reactions where γ-TuRC is absent. This is expected if a significant proportion of the non-branched microtubules are nucleated by γ-TuRCs bound to the glass surface. Importantly, the addition of TPX2 or augmin does not change the number of non-branched microtubules. This means that the increase in branching microtubule nucleation by TPX2 and augmin is not a result of increased microtubule nucleation overall, but specifically by nucleating microtubules from pre-existing microtubules. This is now addressed in the Results section of Figure 3 in the manuscript and it represents a critical point that is also referred to in the Abstract and Discussion.

Second, to further address whether augmin or TPX2 increase the nucleation activity of γ-TuRC, we performed a microtubule nucleation assay in solution as suggested. The results are shown in a new Figure 3—figure supplement 3. Augmin did not enhance the nucleation activity of purified γ-TuRC in vitro consistent with a previous study (Song et al., 2018). We tested TPX2 at the same concentration (50 nM) that was used in the assays described in Figure 3. Unfortunately, at this concentration and in the absence of pre-formed microtubules, TPX2 forms co-condensates with tubulin and nucleates microtubules on its own (King and Petry, 2019). Because these microtubules grow close to each other and are subsequently fixed, it is not possible to count single microtubules. Additionally, quantifying tubulin intensity in this case is not useful because the TPX2 condensates recruit very high levels of tubulin (King and Petry, 2019). To circumvent this hurdle, we tested a lower TPX2 concentration (10 nM), at which TPX2 does not form co-condensates with tubulin. In this case, TPX2 alone did not enhance the nucleation activity of γ-TuRC.

Please note that we are in the process of developing a novel single molecules assay, in which individual γ-TuRCs are attached to the glass surface and can be observed nucleating microtubules live at the single molecule level (Thawani et al., in preparation). While this new assay belongs to its own story, and we could not include it in this *eLife* submission, we also find in this independent assay set-up that TPX2 in solution does not activate γ-TuRC.

These solution experiments, combined with the quantification of de novo microtubule nucleation, suggest that TPX2 and augmin are not mere localization factors and require a pre-existing microtubule to complete this reaction. We address this point in the Results section of these newly added experiments and come back to it in the edited Discussion, as well as the Abstract. Again, we thank you for pointing out this aspect that is critical for this study.

3) It is hard to believe that TPX2 and γ-TuRC, without augmin, can induce branching nucleation, given that TPX2 has no ability to recruit γ-TuRC to existing microtubules (Figure 2—figure supplement 2). The authors show that the branch angle is slightly different from augmin-dependent nucleation (Figure 3F). Is this the only difference? The authors should investigate this peculiar phenomenon more thoroughly.

We thank the reviewers for this comment. First, we repeated the experiment in Figure 2B-C more thoroughly with an important improvement. Before, the background fluorescence level was high, and we hypothesized that this could have masked the detection of γ-TuRC signal when only TPX2 was bound to the microtubule. Indeed, with an improved treatment of the glass surface, the background fluorescence could be reduced. Consequently, we could detect an increase in the level of γ-TuRC signal on microtubules when TPX2 was bound to the microtubule, compared to γ-TuRC alone. The difference is small, but statistically significant and reported in the updated Figure 2C and the respective Results section. This could contribute to the observed levels of branching microtubule nucleation with TPX2 and γ-TuRC alone. Additionally, it is known that TPX2 can recruit tubulin to the lattice of a pre-existing microtubule, potentially making the γ-TuRCs on the microtubule lattice more likely to nucleate new branched microtubules. Last, TPX2 contains γ-TuNA motifs, which have been proposed to activate γ-TuRC. All of these aspects are addressed in the Discussion section. Last, this also leads to a more concrete model of how TPX2 functions in the full reconstitution, which is now brought up in the discussion and nicely summarizes a model of branching microtubule nucleation.

Second, we do not see visual differences between the branched microtubules nucleated by TPX2 + γ-TuRC and by augmin + TPX2 + γ-TuRC, other than the angle distribution. We do see microtubule branches nucleated from distinct TPX2 puncta, similar to when both TPX2 and augmin are present together. This is now shown in the newly incorporated Figure 3—figure supplement 2. Also, we now address in the Discussion how augmin could contribute to a shallower angle distribution.

[Editors' note: further revisions were requested prior to acceptance, as described below.]

All reviewers agree that the revisions have much improved the manuscript, but there are two remaining issues that need to be addressed before acceptance.In the manuscript you argue that TPX2 and augmin are activation factors for γ-TuRC and not just localization factors. The evidence is that TPX2 and augmin do not activate γ-TuRC based nucleation away from the MT lattice, but do increase nucleation of branches from the MT lattice. However, this would actually be the definition of a localization factor, to give more nucleation from a specific location, without requiring activation. When comparing Figure 2C and 3E it seems that there is a stronger increase in branches (Figure 3E) than there is an increase in γ-TuRC on the lattice (Figure 2C). Given the size of the error and the spread of the data, is this a significant difference? At the very least it seems to be a small activation. We would like you to address this point more clearly in the paper. Also, if your conclusion is that the activation is indeed very small, you may want to consider changing the Abstract sentence "TPX2 and augmin do not act as mere localization factors, but enable γ-TuRC-dependent microtubule nucleation at preferred branching angles of less than 90 degrees from regularly-spaced patches along microtubules." to "TPX2 and augmin enable γ-TuRC-dependent microtubule nucleation at preferred branching angles of less than 90 degrees from regularly-spaced patches along microtubules."

We thank the reviewers for prompting us to compare the binding of γ-TuRC versus the branching activity in order to assess whether this supports an activation mechanism or rather enhanced localization.

To evaluate whether the increase in branching microtubule nucleation activity in Figure 3E is larger than the increase in γ-TuRC binding in Figure 2C we calculated the ratio of branching to binding, and normalized it to the γ-TuRC only sample (see Author response table 1 and Author response image 1). The increase in branching when both TPX2 and augmin are present with γ-TuRC is 4x higher than what is expected from the increase in γ-TuRC localization alone. This increase is 3x and 2x higher for TPX2 + γ-TuRC and augmin + γ-TuRC, respectively. However, the standard deviations associated with these ratios indicate relatively high variation in the data, particularly for γ-TuRC only, TPX2 γ-TuRC and augmin + γ-TuRC.

**Author response table 1. resptable1:** 

	Average Binding	Std Dev	Average Branching	Std Dev
gTuRC only	0.21294	0.14638	0.05715	0.04711
TPX2 + gTuRC	0.29349	0.18580	0.23863	0.09174
augmin + gTuRC	0.37321	0.22619	0.20415	0.14939
TPX2 + augmin + gTuRC	0.51905	0.26268	0.56789	0.14644
	Branching / Binding	Normalized Branching/Binding	Std Dev
gTuRC only	0.2684	1.0000	1.0733
TPX2 + gTuRC	0.8131	3.0293	2.2437
augmin + gTuRC	0.5470	2.0381	1.9365
TPX2 + augmin + gTuRC	1.0941	4.0763	2.3154

We thoroughly looked into different tests to assess statistical significance, based on the instruction upon our inquiry on how to best address this issue. However, we could not identify one that would allow us to directly compare ratios of two means from entirely different distributions. This is because each mean, for binding and for branching, has an error associated with it, and the measurements are not paired. While we can propagate this error to get the standard deviation of the ratios, the ratio itself is a single value and there is no sample size associated with it. We inquired with multiple expert sources, including local quantitative and data experts, only to come back to this answer. If the reviewers can suggest a specific statistical test that would be appropriate for this scenario and would work on this sample size, we would very much appreciate it and would gladly calculate the significance.

The only way to apply statistical tests was by ignoring the error of one of the means. To maintain a sample for which we could perform a T-test, we took the individual branching measurements and divided each by the average value of γ-TuRC binding (see Author response table 2).

**Author response table 2. resptable2:** 

	Individual branching measurements	Individual branching measurements divided by the average binding	Average of the individual branching/ binding ratios
γ-TuRC	0	0	0.268392398
0.074895147	0.51719484
0.042944258	0.201673044
0.110766504	0.520177064
TPX2 + γ-TuRC	0.281327868	0.958560319	0.813061147
0.159515074	0.543511104
0.165993471	0.565584759
0.347664851	1.185488405
augmin + γ-TuRC	0.139938427	0.374958943	0.547017648
0.039399551	0.105569387
0.387446726	1.03814669
0.49825122	1.03814669
TPX2 + augmin + γ-TuRC	0.627385371	1.208718565	1.094087931
0.415528909	0.800556611
0.743284806	1.43201003
0.485346276	0.935066548

This procedure yielded four ratios on which we could perform statistical tests. A t-test using these values showed that the difference between γ-TuRC only and augmin + γ-TuRC is not significant at the 5% significance level. This would suggest that augmin does not increase branching microtubule nucleation beyond its increase in γ-TuRC recruitment. In contrast, the differences between γ-TuRC only and TPX2 + γ-TuRC, and γ-TuRC only and augmin + TPX2 + γ-TuRC, are significant at the 5% significance level. Yet, because we are ignoring the error and sample size associated with the average value of γ-TuRC binding, we are not confident to say that TPX2 and TPX2 + augmin are increasing branching microtubule nucleation beyond their increase in γ-TuRC recruitment.

Similarly, we also tried the reverse calculation, where we took individual binding measurements (more numerous because not as challenging, n > 1000) and divided each by the average branching measurement. Because the sample size here remained very large, the t-test showed that all of the differences are statistically significant. However, because we are ignoring the error and sample size associated with the average value of branching microtubule nucleation, we are again not confident to say that TPX2, augmin, or TPX2 + augmin are increasing branching microtubule nucleation beyond their increase in γ-TuRC recruitment.

We also used Z-scores to compare the branching/binding ratios, using the standard deviation from the propagation of error, and a sample size of 1. However, in this case one of the two relevant standard deviations is also ignored (see Author response table 3).

**Author response table 3. resptable3:** 

	Normalized Branching / Binding	Std Dev
γ-TuRC only	0.999963327	1.073272955
TPX2 + γ-TuRC	3.029284911	2.243660565
augmin + γ-TuRC	2.038064177	1.936445753
TPX2 + augmin + γ-TuRC	4.076331128	2.315317404
	Z-scores
	Ignoring the StdDev of gTuRC only	Ignoring the StdDev of the other sample
γ-TuRC only vs TPX2 + γ-TuRC	-0.90446907	Not significant	1.890778645	significant
γ-TuRC only vs augmin + γ-TuRC	-0.536085686	Not significant	0.967229115	Not significant
γ-TuRC only vs TPX2 + augmin + γ-TuRC	-1.328702404	Not significant	2.866342442	significant

For the comparison between γ-TuRC only vs TPX2 + γ-TuRC, and γ-TuRC only vs augmin + TPX2 + γ-TuRC, ignoring the standard deviation of γ-TuRC only shows that they are not significant. Ignoring the other standard deviation yields that the differences are significant. In the case of γ-TuRC only vs augmin + γ-TuRC, ignoring either of the standard deviations yields that the difference are not significant. However, analogous to the t-test, because we are ignoring one of the standard deviations, we are not confident about making a conclusive statement regarding the ability of TPX2 or augmin to increase branching microtubule nucleation beyond their increase in γ-TuRC recruitment.

We believe that the question of whether TPX2 and/or augmin activate γ-TuRC on microtubules to initiate branching microtubule nucleation is very important, but taking into account the information presented above, we do not feel comfortable making a conclusive statement at this point. This is something that requires new experimentation at the single molecule level, and potentially structural insight of the branching factors bound to each other, which is outside the scope of this study. We made sure to clarify and explain our attempt to address this point in the Results section of the manuscript. We also comment on this again in the Discussion. As suggested, we also changed the language in the Abstract to avoid stating that TPX2 / augmin act as more than localization factors. We hope that these efforts adequately addressed your insightful suggestion.

We would also like to ask you to include the data shown in your rebuttal letter as part of Figure 3—figure supplement 2 (GFP-augmin, untagged TPX2 and γ-TuRC).

We thank the reviewers for this suggestion. We agree that it is useful to include this data in the manuscript, and we created a new panel in Figure 3—figure supplement 2 to show this. To confirm, we had performed an additional experiment similar to the one described in Figure 3D with GFP-labeled augmin, but this time using an untagged version of TPX2. We see that augmin is present on the microtubule in puncta similar to the images where both augmin and TPX2 are concurrently labeled with GFP.